

# GPP and the predictability of $CO_2$: more uncertainty in what we predict than how well we predict it

István Dunkl[1,2], Nicole Lovenduski[3], Alessio Collalti[4], Vivek K. Arora[5], Tatiana Ilyina[1], and Victor Brovkin[1,6]

[1]Max Planck Institute for Meteorology, Hamburg, Germany
[2]International Max Planck Research School on Earth System Modelling, Hamburg, Germany
[3]Department of Atmospheric and Oceanic Sciences and Institute of Arctic and Alpine Research, University of Colorado, Boulder, CO, USA
[4]Forest Modelling Lab., Institute for Agriculture and Forestry Systems in the Mediterranean, National Research Council of Italy (CNR-ISAFOM), Perugia, Italy
[5]Canadian Centre for Climate Modelling and Analysis, Environment and Climate Change Canada, University of Victoria, Victoria, British Columbia, Canada
[6]Center for Earth System Research and Sustainability, University of Hamburg, Germany

**Correspondence:** István Dunkl (istvan.dunkl@mpimet.mpg.de)

**Abstract.** The prediction of atmospheric $CO_2$ concentrations is limited by the high interannual variability (IAV) of terrestrial gross primary productivity (GPP). However, there are large uncertainties in the drivers of GPP IAV among Earth system models (ESMs). Here, we evaluate the impact of these uncertainties on the predictability of atmospheric $CO_2$ in six ESMs. We use regression analysis to determine the role of environmental drivers on (i) the patterns of GPP IAV, and (ii) the predictability

of GPP. There are large uncertainties in the spatial distribution of GPP IAV. Although all ESMs agree on the high IAV in the tropics, several ESMs have unique hotspots of GPP IAV. The main driver of GPP IAV is temperature in the ESMs using the Community Land Model, and soil moisture in IPSL-CM6A-LR and MPI-ESM-LR, revealing underlying differences in the source of GPP IAV among ESMs. Between 13% and 24% of the GPP IAV is predictable one year ahead, with four out of six ESMs between 19% and 24%. Up to 32% of the GPP IAV induced by soil moisture is predictable, while only 7% to 13% of the

GPP IAV induced by radiation. The results show that while ESMs are fairly similar in their ability to predict themselves, their predicted contribution to the atmospheric $CO_2$ variability originates from different regions and is caused by different drivers. A higher coherence in atmospheric $CO_2$ predictability could be achieved by reducing uncertainties of GPP sensitivity to soil moisture, and by accurate observational products for GPP IAV.

## 1 Introduction

Near-term predictions of atmospheric $CO_2$ concentrations are an essential step towards the evaluation of climate mitigation efforts and the development of carbon monitoring programs (Ilyina et al., 2021). However, the high interannual variability (IAV) of land-atmosphere carbon fluxes, specifically gross primary productivity (GPP), drives the variability of atmospheric





$CO_2$ and limits its predictability (Piao et al., 2020). The skilful prediction of GPP is therefore a crucial step towards the real-time verification of anthropogenic carbon emissions and the evaluation of mitigation efforts.

The usual approach to evaluate the predictability of an earth system variable is to compare predictions with observed values. In the case of GPP, this is complicated by the uncertainty in GPP observations (Zhang and Ye, 2021). As an alternative to calculating the actual predictability that is based on observations, the potential predictability can be assessed by evaluating how well the models can predict themselves. To do this, an ensemble of simulations with an Earth system model (ESM) is initialized from quasi-identical conditions. In a system with little predictability, the spread among the ensemble members increases

quickly, until the predictive capabilities are lost when the ensemble spread reaches the magnitude of the IAV. There are, however, certain processes in the earth system which provide predictability and hinder the divergence of the ensemble members. For example, the El Niño-Southern Oscillation (ENSO) produces predictable climate anomalies that have a sustained impact on GPP (Zeng et al., 2008; Betts et al., 2016). Other processes extend predictability by providing "memory" that maintains the initial conditions. Soils, for example, store initial moisture anomalies by acting as a buffer between the atmosphere and

the vegetation (Bellucci et al., 2015). Soil moisture anomalies are further extended through land-atmosphere coupling, which creates a feedback loop that enhances the persistence of these anomalies (Kumar et al., 2020). The initial conditions of the simulations are maintained through the lagged response of plant growth to climatic conditions. Slowly reacting vegetation can cause precipitation anomalies or prolonged drought (Alessandri and Navarra, 2008; Zhang et al., 2021). Given all of these mechanisms of predictability, we find that terrestrial carbon fluxes are predictable for two years (Ilyina et al., 2021).

Although several ESMs reproduce the same predictability horizon for globally integrated terrestrial carbon fluxes (Séférian et al., 2018; Ilyina et al., 2021; Spring and Ilyina, 2020; Lovenduski et al., 2019), there are substantial differences in the spatial patterns of GPP IAV (Anav et al., 2015; O'Sullivan et al., 2020). The reason for these differences lies in poorly constrained ecosystem processes that have a large impact on GPP. One of these differences arises from the uncertainty in the sensitivity of GPP to environmental drivers (Ahlström et al., 2015; Jung et al., 2017; Beer et al., 2010; Piao et al., 2020; Collalti et al., 2020).

The sensitivity of GPP to temperature and precipitation varies among studies, leading to the ongoing discussion concerning the dominant driver of global carbon fluxes (Piao et al., 2020). The different sensitivity of GPP to precipitation across ESMs is further exacerbated by the large disagreement in water storage anomalies (Wu et al., 2021). The simulated annual cycle of water storage anomalies of major river basins is between 0.1 and 2 times that of the observed variability. These deviations in hydrological variability between models are likely to cause similar deviations in GPP IAV, especially in semi-arid watersheds.

Further differences in GPP IAV are due to variations in ecosystem boundaries and the related spatial distribution of plant productivity. The Amazon rainforest, for instance, is a hotspot of land-atmosphere carbon fluxes and provides a large contribution to the predictability of atmospheric $CO_2$ (Zeng et al., 2008; Séférian et al., 2018; Ilyina et al., 2021). However, the transition zone between the wet tropical forest and semi-arid tropics within the Amazon basin varies among the models due to differences in their representation of land cover (Collier et al., 2018; Hu et al., 2022). Such differences in biome boundaries also modify

the impact of ENSO on GPP IAV. ENSO produces a distinct spatial pattern of climatic anomalies which influences the GPP on 32% of the vegetated land area significantly (Zhang et al., 2019). These ENSO-related climate patterns will have a different impact on GPP depending on the type of biomes under their influence. In addition to the spatial variability, many ESMs strug-





gle to reproduce the seasonal variability of carbon fluxes. This can be seen in the large biases in phenology (Song et al., 2021). Several models overestimate the seasonal amplitude of leaf area index (LAI) in the tropics, and mismatch the timing of LAI

maxima and minima (Peano et al., 2019, 2021).

  All of these uncertainties suggest, that there are large differences in the patterns of GPP IAV among the ESMs. With this study, we want to extend our understanding of GPP predictability by considering the different patterns of GPP IAV among the ESMs as well. In a multi-model analysis, we investigate which processes drive the IAV of GPP and which processes allow the GPP IAV to be predictable. Regression analysis is used to determine the role of three environmental variables (soil moisture,

temperature, and radiation) on GPP IAV and GPP predictability. We analyse the cause of differences in GPP predictability across ESMs, identify the areas of large discrepancies, and determine the factors contributing to the attached uncertainties. The aim of this study is to reveal which factors of GPP representation are limiting the predictability of atmospheric $CO_2$.

## 2 Methods

### 2.1 Data sources

We analyse model output from the Decadal Climate Prediction Project (DCPP, boer2016decadal). This protocol-driven multi-model approach aims at studying the decadal predictability of the earth system with hindcasts, quasi-real-time forecasts, and case studies on predictability mechanisms. The hindcasts are initialized annually from 1960 to 2017 or 2019 with the starting dates between November and January and at least 10 ensemble members. Simulations are driven by CMIP5 or CMIP6 historical forcing and extended by RCP4.5 or SSP2-4.5 afterwards. The DCPP framework does not prescribe any specific initialization

or data assimilation methods and leaves these details to be decided by the respective modelling centres.

  We additionally use the CESM2 model output from the Seasonal-to-Multiyear Large Ensemble (SMYLE) prediction system (Yeager et al., 2022). The SMYLE hindcasts ensembles are initialized four times per year with 20 ensemble members between 1970 and 2019. In this study, the November initializations are used to achieve the highest comparability with the DCPP hindcasts.

We compare the spatial GPP IAV patterns of the ESMs with observation-based GPP products. Because of the uncertainty among observations, we include products based on three different sources. The Vegetation Photosynthesis Model (VPM, Zhang et al. 2017) is remote sensing-based product that uses a light use efficiency (LUE) model to calculate GPP. VPD uses satellite data from MODIS and an improved LUE algorithm that considers leaf quality. The second data set is GOSIF (Li and Xiao, 2019) which is based on data from MODIS and the Orbiting Carbon Observatory-2. GOSIF uses solar-induced chlorophyll

fluorescence, which is a more recent approach to calculate GPP. Lastly, we use FLUXCOM (version RS + METEO (ERA5), Jung et al. 2019), which uses machine learning to upscale flux tower observations with meteorological and remote sensing data. Because FLUXCOM underestimates the IAV of GPP (Anav et al., 2015; O'Sullivan et al., 2020), it is recommended to scale the data so that the IAV of its integrated fluxes resembles observations (Jung et al., 2019). The VPM, GOSIF, and FLUXCOM data are linearly detrended before calculating the IAV. Due to the long time span, FLUXCOM is detrended over two periods

(1979 to 1999, and 2000 to 2018).



**CanESM5**

The Canadian Earth System Model version 5 (CanESM5; Swart et al. 2019) consists of the Canadian Land Surface Scheme (CLASS) and Canadian Terrestrial Ecosystem Model (CTEM) with a T63 grid of an approximate resolution of 2.8°. The atmosphere is realized with the Canadian Atmospheric Model (CanAM5) with 49 vertical levels. Ocean physics is simulated with CanNEMO, on a tripolar grid with a resolution of 1° to 1/3° and 45 vertical levels, and ocean biogeochemistry is represented by the Canadian Model of Ocean Carbon (CMOC).

The CanESM5 hindcast simulations are initialized every January between 1960 and 2017 with 20 members. 3D potential temperature and salinity of the global oceans are nudged toward monthly Ocean Reanalysis System 5 (ORAS5; Zuo et al. 2019). Sea surface temperatures are nudged to Extended Reconstructed Sea Surface Temperature (ERSSTv3; Xue et al. (2003); Smith et al. (2008)) until 1981 and to Optimum Interpolation Sea Surface Temperature (OISST; Banzon et al. 2016) afterwards. Sea ice concentration is nudged to the Hadley Centre Sea Ice and Sea Surface Temperature data set (HadISST.2; Titchner and Rayner 2014), and sea ice thickness to monthly climatology until 1988 and to the SMv3 statistical model of Dirkson et al. (2017) afterwards. For the atmosphere, temperature, horizontal wind components and specific humidity are nudged to ERA40 (Uppala et al., 2005) until 1978 and to 6-hourly ERA-Interim data (Dee et al., 2011) afterwards.

**CESM1-CAM5**

The Community Earth System Model (CESM) version 1.1 (Hurrell et al., 2013) is used to produce 40-member simulations in the Decadal Prediction Large Ensemble (DPLE) project (Yeager et al., 2018). The model components are the Community Land Model version 4 (CLM4; Lawrence et al. 2011) with a 1° resolution, Community Atmosphere Model Version 5 (CAM5) with 30 vertical levels, the Parallel Ocean Program (POP2) with 60 vertical levels and sea ice with Community Ice Code (CICE4).

The CESM1-CAM5 hindcasts are initialized every November. There is no direct assimilation of observations to produce the initial conditions, instead, ocean and sea ice are obtained from simulation runs forced by historic atmospheric surface fields (Yeager et al., 2018). Initial conditions for the land and atmosphere components are obtained from ensemble member #34 of the CESM Large Ensemble (Kay et al., 2015; Lovenduski et al., 2019).

**CESM2**

CESM version 2 (Danabasoglu et al., 2020) runs on a 1° horizontal resolution of all components. The atmosphere is simulated by the Community Atmosphere Model Version 6 (CAM6) with 32 vertical levels. The ocean model is the Parallel Ocean Program version 2 (POP2) with 60 vertical levels, with the biogeochemistry from the Marine Biogeochemistry Library and sea ice by CICE version 5.1.2 (CICE5) with 8 vertical layers. The land component is simulated by the Community Land Model version 5 (CLM5; Lawrence et al. 2019), which has several updates to its predecessor CLM4 and CLM4.5, leading to a better representation of the global carbon cycle in benchmarks (Bonan et al., 2019).

Hindcasts are initialized on the 1st of every November, February, May and August, and run for 24 months. Only the November initializations are used in this analysis to increase comparability with the DCPP simulations. Initial conditions for the



atmosphere, ocean and sea-ice stem from the Japanese 55-year Reanalysis (JRA-55; Kobayashi et al. (2015), and JRA55do; Tsujino et al. (2018)). The land surface and biogeochemistry are initialized from forced CLM5 simulations.

**CMCC-CM2-SR5**

The Euro-Mediterranean Centre on Climate Change coupled climate model (CMCC-CM2, Cherchi et al. (2019), Lovato et al. (2022)) is based on CESM and consists of the Community Land Model (CLM4.5) with a 1° resolution, the atmospheric model CAM5.3 with 30 vertical levels. The distinguishing element of CMCC-CM is the ocean, which is simulated by NEMO3.6, while sea ice is modelled by CICE4.

The 10-member hindcast simulations are initialized every November (Nicolì et al., 2022). The ocean initial conditions are from CHOR (Yang et al., 2017) until 2010 and from CGLORSv7 (Storto and Masina, 2016) afterwards. The atmosphere is initialized from ERA-40 until 1978 and by ERA-Interim afterwards. The land surface is initialized using the reanalysis with two different meteorological forcing. For this reason, only the ensemble members 1, 3, 5, 7, 8 and 9 are used, as the other members start from a different state and this would not allow quantifying predictability by ensemble spread.

Because the CMCC-CM2-SR5 fields containing land-atmosphere carbon fluxes are not exported for the DCPP runs, the historical simulations are used to infer the relationship between environmental drivers and GPP.

**IPSL-CM6A-LR**

The ESM developed by the Institute Pierre Simon Laplace (IPSL; Boucher et al. 2020) uses the ORCHIDEE v2.0 (Cheruy et al., 2020) land surface model (LSM) with an average resolution of 157 km. The atmosphere is simulated at the same resolution by LMDZ6 with 79 vertical levels, the ocean with NEMO-OPA with a 1° resolution and 75 vertical levels and ocean biogeochemistry with PISCESv2.

The hindcast simulations of IPSL-CM6A-LR come from the DCPP project. The 10-member ensembles start annually in January between 1960 and 2016. The hindcasts are initiated from an assimilation run with EN4 sea surface temperatures (Good et al., 2013) and Atlantic sea surface salinity (Estella-Perez et al., 2020). Subsurface ocean, sea ice and atmosphere are not assimilated.

**MPI-ESM-LR**

MPI-ESM-LR is the Max Planck Earth System Model (MPI-ESM1.1; Giorgetta et al. 2013) used in a low-resolution configuration. The land is simulated by JSBACH with dynamic vegetation (Reick et al., 2013). The ocean component is MPIOM with a horizontal resolution of about 150 km and 40 vertical levels. The atmosphere is simulated by ECHAM at a T63 resolution with 47 vertical layers, and ocean biogeochemistry is represented by HAMOCC.

The utilized hindcast simulations of MPI-ESM-LR are conducted within the MiKlip project (Marotzke et al., 2016). The decadal prediction system are 10-member ensembles starting every January between 1961 and 2014. Ocean temperature and




salinity are initialized from the Ocean Reanalysis System 4 (ORAS4; Balmaseda et al. 2013) and the atmosphere by ERA-40 from 1960 to 1998 and ERA-Interim from 1990 to 2014.

## 2.2 Statistical approach

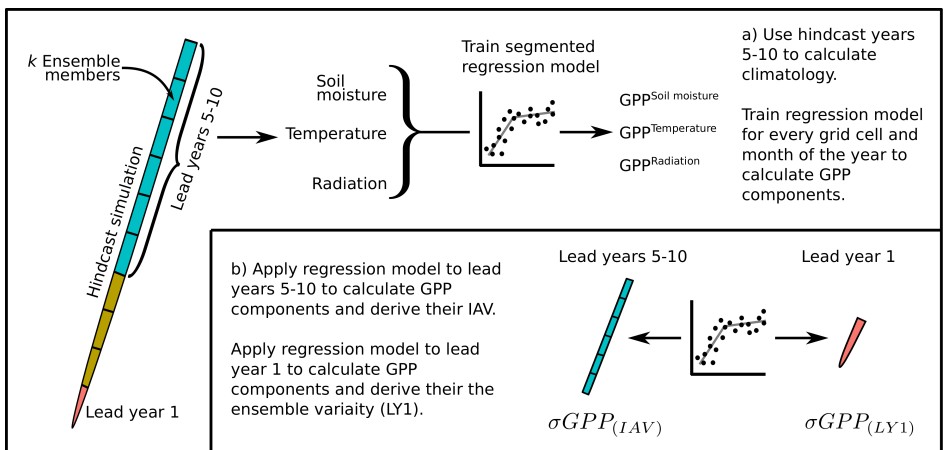

**Figure 1.** Workflow of the statistical analysis: a) Lead years five to ten of the hindcast simulations are used to train a regression model that calculates the components of GPP caused by the environmental drivers. b) The regression model is applied to (i) lead years five to ten to calculate the IAV of the GPP components ($\sigma GPP_{(IAV)}$) and (ii) to lead year one to calculate the mean ensemble spread of the GPP components ($\sigma GPP_{(LY1)}$).

### Overview

An overview of the statistical analysis is shown in Figure 1. Every hindcast simulation is initialized from quasi-identical conditions. With the increasing lead-time, the variability within the hindcast ensemble (standard deviation across the ensemble members for a given time) increases too, until it reaches the IAV. Based on this assumption, the hindcast simulations are split

into two groups by lead-time (lead year one, and lead years five to ten). For the lead years five to ten, the effects of initialization are assumed to be negligible. These years are used to calculate the monthly mean climatology, which is removed from both groups to obtain the anomalies. The anomalies of the lead years five to ten are used in a regression analysis to derive the sensitivity of GPP to the environmental drivers i.e. soil moisture, temperature and radiation (Fig. 1 a)). The regression model is applied to (i) the anomalies of lead years five to ten to calculate the IAV of all GPP components ($\sigma GPP_{(IAV)}$), and (ii) to the

anomalies of lead year one to calculate the ensemble variability of all GPP components ($\sigma GPP_{(LY1)}$, Fig 1 b)). We derive the predictability of GPP by comparing $\sigma GPP_{(LY1)}$ to $\sigma GPP_{(IAV)}$. Because the hindcast simulations are not evaluated against observations, the calculated predictability reflects the potential predictability.



**Climatology and sensitivity**

The monthly mean climatologies are calculated from the lead years five to ten, with a three-year moving window approach
for every calendar year. Because the moving window method is not applicable for the first decade of hindcast initializations,
the monthly climatology for the 1960s (1970s for CESM2) is calculated based on all lead years five to ten within the 1960s
(or 1970s). Anomalies of all input fields are calculated by subtracting the monthly climatologies from the hindcast data. The
obtained anomalies of lead years five to ten make up a data set of $n$ simulation years:

$$n = 6 \; hindcast \; years \; \times \; No. \; ensemble \; members \; \times \; No. \; initializations. \tag{1}$$

With 10 to 40 ensemble members and 56 to 58 initializations resulting in sample sizes of 3330 to 13680. Because the hindcast
length is only two years for the CESM2 simulations, a different approach is used here. Instead of lead years five to ten, only
lead year two is selected and only five random ensemble members are used from every hindcast to reduce the number of
simulations with the same initial conditions. To offset the reduced number of data points, five random simulations are added
from the hindcast simulations initialized in February, May, and August as well.

The resulting data set of lead year 5 to ten anomalies is used to derive the sensitivity of GPP to the environmental drivers
($ENV$: soil moisture, temperature and radiation) by fitting a regression model for every grid cell and month of the year. The
relationship between GPP and the environmental drivers is frequently non-linear, sometimes due to specific break points in
the functional representation of GPP. For this reason, segmented linear regression (SLR) is used to model GPP from the envi-
ronmental drivers (Muggeo, 2008). SLR finds breakpoints in the data, splitting it into multiple ranges and fitting an individual
regression model to each of the data ranges. Here, a single breakpoint is determined for each of the three predictor variables.

    Because environmental drivers have some degree of collinearity, the regression analysis will not be able to fully attribute
the GPP anomalies to their specific causes. Therefore, the resulting sensitivities should be taken as a "contributive", and not a
"true" effect of the environmental drivers (Wang et al., 2016).

**Variability and predictability**

The SLR can now be applied to individual simulations to determine the component of GPP anomalies that can be attributed to
each of the environmental drivers:

$$\Delta GPP \approx \Delta GPP^{Soil \; moisture} + \Delta GPP^{Temerature} + \Delta GPP^{Radiation}. \tag{2}$$

The three components of GPP anomalies ($\Delta GPP^{ENV}$) are calculated for every simulation within the hindcast lead time five
to ten. From the results, we calculate the IAV of the components for every grid cell and month of the year ($\sigma GPP_{(IAV)}^{ENV}$).
Similarly, the SLR is applied to the anomalies of lead year one, to calculate the standard deviation for every month within the
hindcast simulations. Averaging over the standard deviations of every hindcast returns the ensemble variability of lead year
one ($\sigma GPP_{(LY1)}^{ENV}$).

    The predictability is assessed by comparing $\sigma GPP_{(LY1)}$ to $\sigma GPP_{(IAV)}$. A high predictability of an input field means, that
the ensemble variability is restricted for some time after the hindcast initialization, and does not reach the IAV immediately.



To take different aspects of predictability into account, we calculate a relative and an absolute predictability metric. As the absolute metric, we calculate the predictable component $pc$. It is the difference between IAV and ensemble variability (Fig. 2):

$$pc^{ENV} = \sigma GPP_{(IAV)}^{ENV} - \sigma GPP_{(LY1)}^{ENV}. \tag{3}$$

The $pc$ can be interpreted as the amount of IAV that is predictable during lead year one. It allows quantifying the regional contribution to atmospheric $CO_2$ predictability. However, $pc$ is strongly related to the overall magnitude of GPP IAV and does not give us an insight into how well memory is stored in the system. Therefore, the predictable fraction $pf$ is calculated as the fraction of $pc$ to IAV:

$$pf^{ENV} = \frac{pc^{ENV}}{\sigma GPP_{(IAV)}^{ENV}}. \tag{4}$$

$pf$ is a relative metric and allows assessing how well memory is retained in the system.

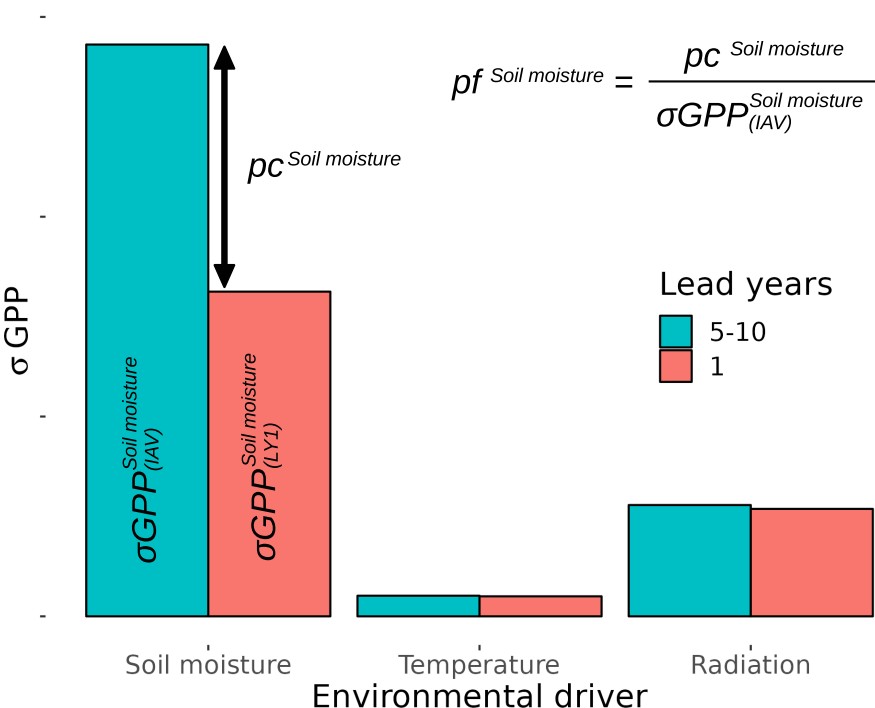

**Figure 2.** The exemplary composition of GPP variability and the calculation of the predictability metrics. The components of GPP IAV are calculated from the lead years five to ten (green bars), and the ensemble variability is calculated from lead year one (red bars). The predictable component ($pc$) results from the difference between IAV and ensemble variability. The predictable fraction ($pf$) is the fraction of $pc$ to IAV. In the exemplified region, most of the variability is caused by soil moisture and radiation, while GPP is not restricted by temperature. Predictability is exclusively provided through soil moisture.



## 3   Results and Discussion

### 3.1   Patterns of GPP IAV

In order to understand what the models are predicting, we start by analysing the patterns of GPP IAV. There are differences in the overall magnitude of GPP IAV among ESMs with CanESM5, CMCC-CM2-SR5, and IPSL-CM6A-LR at the lower, and CESM2 and MPI-ESM-LR at the higher end of the IAV spectrum. Factors that could explain some of the differences in the overall magnitude of IAV is the relatively weak ENSO teleconnection in CanESM5 (Swart et al., 2019), or the low total GPP in CMCC-CM2-SR5 (Lovato et al., 2022).

Because we focus on the spatial patterns of IAV rather than absolute differences, the GPP IAV patterns are scaled for better comparison (Fig. 3). We find agreement in the large-scale patterns of GPP IAV, with most of the IAV of the ESMs in the northern Amazon Basin, and the semi-arid tropics like South America, Africa, South Asia, Australia, and southern North America. A closer examination of GPP IAV reveals that the ESMs have less agreement in the regions contributing most to the IAV, especially in the semi-arid tropics. Some ESMs have large hotspots of GPP IAV, which cannot be found in other ESMs. These unique hotspots are the western Amazon Basin (CanESM5), central South America (CESM2), Southern Africa (MPI-ESM-LR and CanESM5), and Australia (MPI-ESM-LR). We can find the most consistency on the northern coast of South America, which is a high IAV region in most ESMs. The spatial patterns of IAV have an average correlation of 0.47 among the ESMs. The ESM with the lowest correlation values is CESM2 with an average of 0.29. CESM2 stands out with very low IAV in the tropical rainforests of the Amazon and Congo Basins and Southeast Asia.

The correlation among the observational products is 0.65, and although they confirm most of the IAV patterns of the ESMs we find stronger deviations in South America. While many ESMs have IAV hotspots along the northern coast of South America, this is only reproduced in FLUXCOM. However, all observational products show a high GPP IAV in western South America, which can not be found in the ESMs. The spatial patterns of GPP IAV revealed here correspond with the literature, which suggests that the semi-arid tropics, tropical forests, grasslands and croplands are the main drivers of global GPP IAV (Ahlström et al., 2015; Piao et al., 2020; O'Sullivan et al., 2020). These studies also reflect the large uncertainty in the contribution of the individual semi-arid regions to GPP IAV between the models, and in particular the uncertain role of Australia. In an ensemble of eight LSMs, Australia contributed 39%, semi-arid tropical Africa 32%, and Southeast Asia 10% to global GPP IAV, while temperate South America only contributed 2% (Chen et al., 2017). Although Australia has the highest mean model IAV, the variability of IAV between the models is also the largest, with the IAV of GPP ranging between 0.26 and 1.01 Pg C yr$^{-1}$. However, ESMs are likely to underestimate the role of tropical forests in GPP IAV, due to a misrepresentation of photosynthesis (O'Sullivan et al., 2020). In this study, this is especially evident for CESM2, where GPP IAV increases abruptly outside the boundaries of tropical forests.

The divergence in GPP IAV across different ESMs is largely caused by three factors: the sensitivity of carbon fluxes to climatic drivers (Piao et al., 2020) (discussed in section 3.2), phenology (Chen et al., 2017; Peano et al., 2019, 2021), and meteorological input (Anav et al., 2015). The role of phenology is crucial because the amount and quality of leaves determine the exchange of water, $CO_2$ and energy between the land and the atmosphere (Peano et al., 2021). Most LSMs tend to have



**Figure 3.** GPP IAV in three observational products (VPM, GOSIF, and FLUXCOM) and six ESMs. a) Spatial patterns of GPP IAV with brighter colours standing for higher values. The data is scaled across ESMs to highlight differences in patterns and not absolute differences. b) The spatial correlations between the products.

a better representation of the growing season type, and growing season boundaries in the wet than in the semi-arid tropics. Peano et al. (2021) analysed the start- and ending months of growing seasons (GSS and GSE) in eight LSMs under the same





climate forcing and found several regions with a wide range of simulated growing seasons. The largest uncertainties in GSS and GSE are in the semi-arid tropics, the same regions where we find little agreement in GPP IAV. GSS ranges from February to October in Australia, and from March to October in Southern Africa, while GSE ranged from March to September in Africa between 0 and 15°N. The vegetation types with the largest uncertainty in growing season timing are broadleaf deciduous shrubs, which are mostly located in Northern Australia, Southern Hemisphere crops, broadleaf evergreen trees and grasses.

The better-performing models have a high number of plant functional types, or more complex phenology schemes, while the difficulties in semi-arid regions originate from the response of photosynthesis to soil moisture. Its complex phenological scheme puts ORCHIDEE among the better-performing LSMs, and might explain the high correlation of IPSL-CM6A-LR with the GPP IAV of all three observational products.

  It is also a misrepresentation of phenology that can explain the overall high IAV in MPI-ESM-LR. JSBACH overestimates

the seasonality of LAI in the tropics, as it becomes visible in the strong seasonal cycle of tropical LAI in MPI-ESM-LR (Song et al., 2021). Consequentially, the area of the evergreen tropics is underestimated in JSBACH (Peano et al., 2021). This leads to a larger fraction of semi-arid tropics with a higher IAV. This amplification of the equatorial dry season might lead to the high GPP IAV in the Northern Amazon and contribute to the overall high IAV in MPI-ESM-LR (Wang et al., 2011).

  To examine the role of meteorological input on the differences in GPP IAV among ESMs, we compare soil moisture IAV

(Figure 4) with GPP IAV. Notable is the relationship between soil moisture and GPP IAV in Australia and Southern Africa. MPI-ESL-LR and IPSL-CM6A-LR which have the highest GPP IAV in Australia, have low soil moisture IAV in this region compared to the CLM family. We see similar results for Southern Africa, where all models but CMCC-CM2-SR5 and IPSL-CM6A-LR have relatively high soil moisture IAV, however, GPP IAV is low in the CLM family and high in IPSL-CM6A-LR. This disagreement between the variability of GPP and its forcing is in accordance with the findings of Peano et al. (2021),

supporting the assumption that the response of GPP to soil moisture is not well constrained in semi-arid ecosystems. The high soil moisture variability in CESM2 in Australia and Southern Africa despite low GPP IAV could result from its low climate sensitivity (Wieder et al., 2021).

## 3.2   Drivers of GPP IAV

  To determine the drivers of GPP IAV, we analysed the sensitivity of GPP to environmental drivers using regression analysis.

The globally averaged contribution of the drivers to GPP IAV is shown as the bars in Figure 5. The CLM family and CanESM5 have similar patterns, with temperature dominating the IAV, or being on par with soil moisture. IPSL-CM6A-LR and MPI-EMS-LR have a distinctly different pattern, with soil moisture dominating the IAV and radiation contributing equally or more than temperature. A reason for the large contribution of soil moisture to GPP IAV in IPSL-CM6A-LR and MPI-ESM-LR could be that both ESMs are at the high end of soil moisture IAV for deep soil layers in the Southern Hemisphere (Qiao et al., 2022),

where many of the semi-arid ecosystems are located that contribute most to GPP IAV. Another explanation could be that out of eleven ESMs, IPSL-CM6A-LR and MPI-ESM-LR have the lowest warm-season soil moisture (Padrón et al., 2022). This increase in dryness can lead to a larger extent of semi-arid ecosystems with a generally higher GPP IAV. Another effect of the reduced warm-season soil moisture can be an increase in land-atmosphere coupling strength (Santanello et al., 2018). This





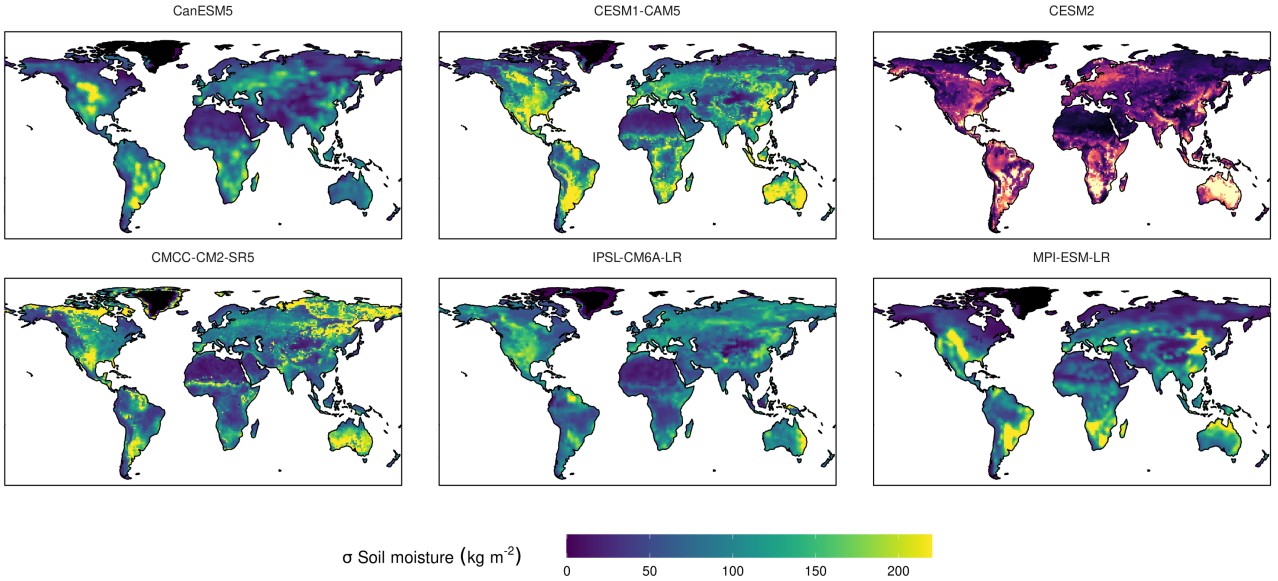

**Figure 4.** IAV of soil moisture in 6 ESMs. Note that CESM2 calculates soil moisture over a deeper soil column, hence the variability of CESM2 is scaled to a similar range as the other models.

would explain the higher correlation between soil moisture and temperature in IPSL-CM6A-LR and MPI-ESM-LR (Padrón
et al., 2022), and make the regression coefficients shift towards the stronger predictor, which is soil moisture.

The spatial drivers of GPP IAV show agreement in the wet and arid tropics, while there is little consistency in the semi-arid transition zones (Fig. 5). In many ESMs, the GPP IAV in the wet tropics, and eastern China is induced by radiation, while soil moisture becomes more prevalent along the aridity gradient, and is driving IAV in Southern Africa, Southern South America, and Australia. The IAV of the remaining land surface is driven predominantly by soil moisture in IPSL-CM6A-LR
and MPI-ESM-LR, and by a combination of temperature and soil moisture in the remaining ESMs.

Multi-model averages and observations of GPP sensitivity agree with the larger role of temperature in tropical forests, radiation in western Amazonia, and the importance of precipitation in the semi-arid tropics (O'Sullivan et al., 2020; Anav et al., 2015). However, the role of water on carbon fluxes increases when soil moisture is used instead of precipitation in sensitivity studies (Piao et al., 2020). This can be observed in the sensitivity of net biome productivity (NBP), showing a
more balanced contribution of soil moisture and temperature in the wet tropical forests (Piao et al., 2020; Padrón et al., 2022). Although the comparison of GPP and NBP imposes limitations, GPP explains the majority of tropical NBP (Ahlström et al., 2015). This suggests, that the low water sensitivity of tropical GPP might explain the lower than expected GPP IAV of tropical forests in ESMs.





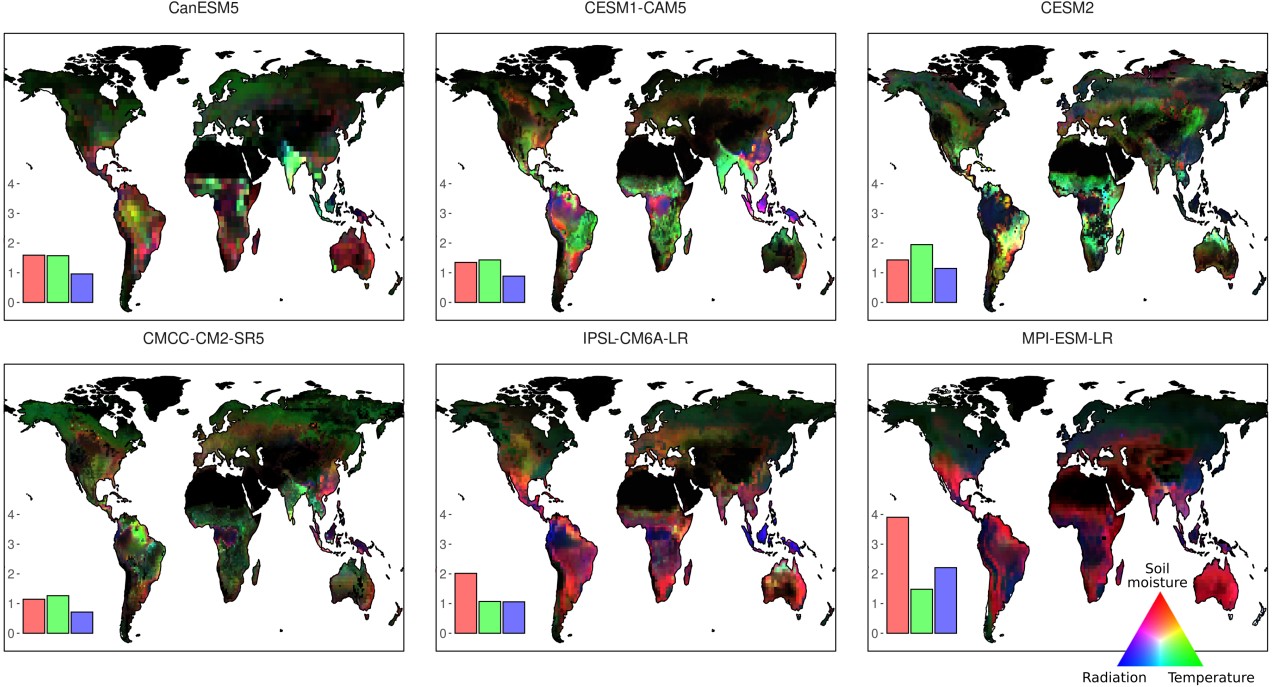

**Figure 5.** The contribution of environmental drivers to GPP variability ($\sigma GPP^{ENV}_{(IAV)}$). Colour intensity stands for higher GPP IAV. The data is scaled across ESMs to highlight differences in patterns and not absolute differences. Bars represent the mean contribution of environmental drivers to global GPP IAV (kg C m$^{-2}$ s$^{-1}$ 10$^{-13}$).

### 3.3 Predictability of GPP

To analyse the role of GPP in the predictability of atmospheric CO$_2$, we assessed GPP predictability using two metrics. The predictable component ($pc$) is calculated as the difference between ensemble variability and IAV and provides a measure of absolute predictable variability. $pc$ can be used to assess the predictability of GPP fluxes that contribute to CO$_2$ variability. The predictable fraction ($pf$) is the ratio of $pc$ to IAV and illustrates how well memory is retained in the system. This metric can be used to compare the predictive performance of different biomes, for example.

There is relatively high consistency among the $pf$ of the environmental drivers across the models ($pf^{Soil\,moisture} > pf^{Temperature} > pf^{Radiation}$, numbers above the bars in Fig. 6). This pattern reflects the anticipated differences in predictability among the drivers. Atmospheric fields, as radiation, have a low persistence, leading to a low predictability of two weeks for most regions (Zeng et al., 2008). Soil hydrology, on the other hand, acts as a low-pass filter which removes the unpredictable high-frequency variability of precipitation and allows a predictability of soil moisture of around two years (Chikamoto et al., 2017). Temperature gains most of its predictability through sea surface temperature (SST) forcing in the equatorial regions (Feng et al., 2011), and land-atmosphere coupling in the semi-arid tropics (Seo et al., 2019).



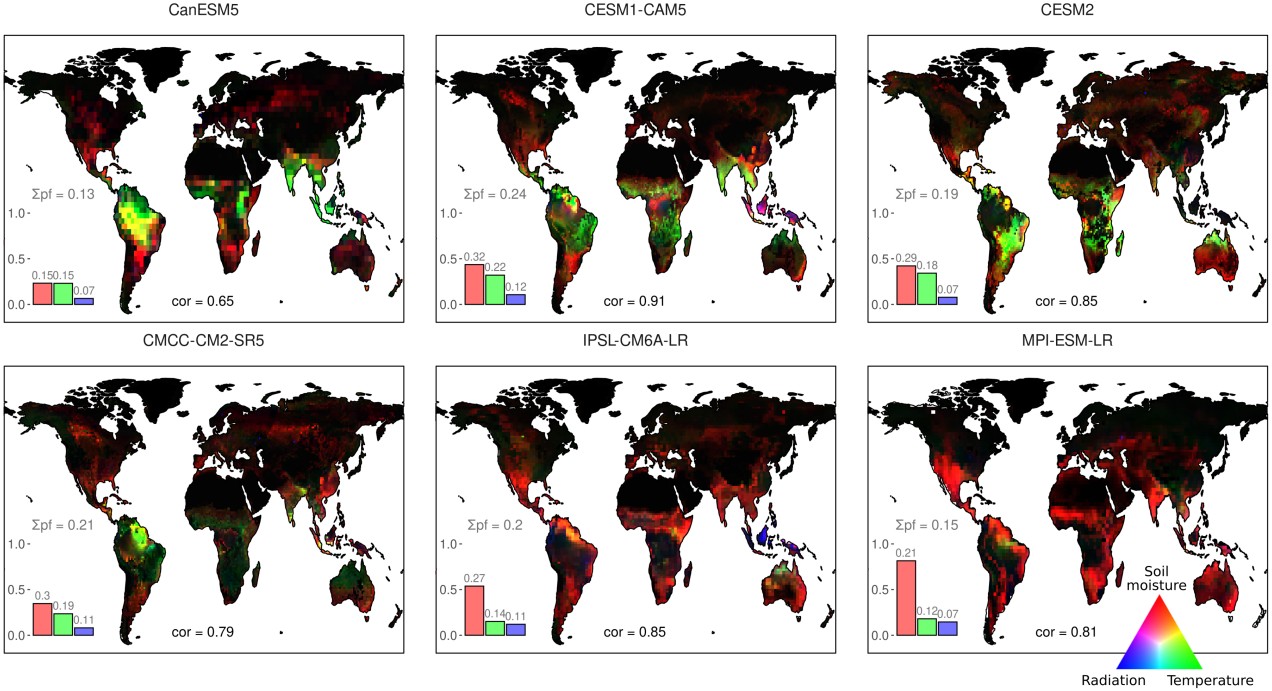

**Figure 6.** The contribution of environmental drivers to the predictable component of GPP ($pc$). The contribution is calculated as the difference between the IAV and the ensemble variability within lead year one of the hindcast experiments. Values are scaled for each ESM. Bars represent the mean contribution of environmental drivers to the $pc$ ($\Delta\sigma$ GPP in kg C m$^{-2}$ s$^{-1}$ 10$^{-13}$). Numbers on top of the bars show the predictable fraction ($pf$), which is the share of the $pc$ to overall IAV. The correlation between GPP IAV and the $pc$ is shown at the bottom of the plots.

The overall $pf$ of CESM2, CMCC-CM2-SR5 and IPSL-CM6A-LR falls into a narrow window of 0.19 to 0.21. CESM1-CAM5 has the highest $pf$ with a value of 0.24. It is likely that this increased share of predictable IAV is not due to differences in model structure, but due to the large number of 40 ensemble members. Most other ESMs in this study have only ten

ensemble members, which is not enough to capture the difference between ensemble variance and IAV, so that an increase in ensemble members leads to an increase in prediction skill (Meehl et al., 2021). However, despite having 20 ensemble members, CanESM5 has the lowest $pf$ among the models. A possible explanation could be its poor representation of soil moisture (Qiao et al., 2022). In particular, the low variability of deep soil moisture in CanESM5 could hinder the persistence of the initial climatic conditions. On the other hand, a high variability of soil moisture does not guarantee a high $pf$, as seen by the example

of MPI-ESM-LR. The low $pf$ of MPI-ESM-LR can be explained by the sensitivity of GPP to radiation. Since only 7% to 12% of the radiation-induced IAV is predictable, a high share of $\sigma$GPP$^{Radiation}$ reduces the predictability of GPP. This becomes evident in MPI-ESM-LR, in which the share of $\sigma$GPP$^{Radiation}$ is 20% higher than in the other ESMs.




We find that the regions contributing to the predictability of atmospheric $CO_2$ ($pc$) are highly related to the IAV patterns. The spatial correlation between $pc$ and IAV exceeds 0.79 in all models but CanESM5. Indeed, these high correlations between predictability and IAV align with our understanding. Under a constant $pf$, $pc$ would grow linearly with increasing IAV, leading to a perfect correlation. These high correlations show that the differences in the predictability of atmospheric $CO_2$ are determined more by the differences in GPP IAV than the differences in the $pf$ of GPP. While the $pf$ values show that the ESMs have a similar degree of memory retention, there are few overlaps in the spatial distribution of the $pc$, with an average correlation of 0.38 between the ESMs. For an alternative quantification of this disagreement, we separate the high-predictability grid cells, which are the grid cells contributing to the top 20th quantile of $pc$. 74% of these high-predictability grid cells are unique to only one ESM, and only 8% of high-predictability grid cells can be found in three or more ESMs.

Although the spatial patterns of the $pc$ broadly resemble the patterns of GPP IAV, there are some slight differences between these fields. The $pc$ is relatively high along the northeastern coast of South America in most ESMs. This could be due to the high climate predictability caused by slowly evolving Atlantic SST patterns (Dirmeyer et al., 2018). Other systematic differences can be explained by the differing $pc$ of the environmental drivers. The most evident is the difference between IAV and predictability in regions where GPP IAV is driven by radiation. This leads to relatively low predictability in the tropical rainforests of the western Amazon basin and the Congo basin. An exception is the predictability provided through radiation on the Southeast Asian islands in IPSL-CM6A-LR and CESM1-CAM5. High predictability in these regions could be explained by the proximity to the ENSO SST region. Strong and predictable SST anomalies in the tropical Pacific that surround the islands can directly influence the cloud cover over land. The predictable component is also higher over areas where IAV is driven by soil moisture rather than temperature. In many ESMs, this leads to a high predictable component in the semiarid regions of South America, Africa and India.

## 4 Conclusions

We tested the ability of six ESMs to predict terrestrial GPP and determined the similarities and sources of uncertainties. The ESMs are fairly similar in their ability to retain memory in hindcast simulations, with the $pf$ values of four of the ESMs falling between 19% and 24%. Most of the GPP $pf$ is provided by soil moisture. Up to 32% of the GPP IAV caused by soil moisture is predictable, while it is only 7% to 12% for the IAV caused by radiation. The differences in the $pf$ among ESMs are due to ensemble size and the sensitivity of GPP to radiation. Further sources of predictability which is not studied here are long-term vegetation dynamics. Specifically, the large and structural changes like tree mortality (Wigneron et al., 2020) and recruitment (Holmgren et al., 2001). These processes only occur in extreme years and cause shifts in ecosystem states with long-lasting effects. The correct representation of these processes in ESMs allows them to reproduce the low-frequency IAV of vegetation, thereby extending the $pf$ of GPP.

Although ESMs are similar in the fraction of GPP IAV they can predict, there are substantial differences in the patterns and drivers of GPP IAV. The ESMs have distinct, non-overlapping hotspots of GPP IAV that drive the variability of atmospheric $CO_2$. We find large disparities in the role of Australia, Southern Africa, and central South America on GPP IAV. The leading



cause of the uncertainties in IAV patterns are differences in the response of GPP to soil moisture and the capability of the ESMs to simulate soil hydrology accurately. These differences materialize through the direct effect of soil moisture on photosynthesis, and through the role of soil moisture on phenology. The inability of ESMs to reproduce GPP IAV also means, that there are regions where the potential predictability of GPP does not resemble the actual predictive skill.

These mismatches in GPP IAV imply, that the IAV of atmospheric $CO_2$ are caused by different regions and by different drivers across the ESMs. Consequentially, when ESMs predict the atmospheric $CO_2$, their predictions are originating from different regions. Therefore, the prediction of atmospheric $CO_2$ relies less on the prediction of regional climate anomalies, and more on the predictable global climate patterns like ENSO. These global climate anomalies are able to balance out the regional differences in GPP patterns, by affecting large parts of the land surface simultaneously. In order to utilize the benefits of regional

climate predictability for the predictability of $CO_2$, further work ought to focus on constraining carbon flux variability, and not on the processes providing predictability. While the distribution of mean GPP has long been used to parameterize ESMs, effort should also be directed towards tuning the IAV of GPP to increase the predictive performance of ESMs. A missing piece towards this goal, however, is a robust and independent product of global GPP IAV.

*Code and data availability.* The data and code to produce the figures shown in this study are available at [TBD]

*Author contributions.* The Study was conceptualised by ID, TI, and VB, ID developed the methodology, ran the analysis, and wrote the original draft, NL, AC, VA, TI and VB reviewed and edited the manuscript, and VB and TI supervised.

*Competing interests.* The authors declare that they have no conflict of interest.

*Acknowledgements.* We acknowledge funding from European Union's Horizon 2020 research and innovation programme under Grant Agreement No. 821003 'Climate-Carbon Interactions in the Coming Century CCiCC (4C)'. NL is grateful for funding from the US National Sci-
ence Foundation (OCE-1752724). The authors wish to thank the modelling groups who have participated in the DCPP, SMYLE and MiKlip projects. We also thank Jürgen Bader for reviewing the manuscript.



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
