# Peer review of "GPP and the predictability of $CO_2$ : more uncertainty in what we predict than how well we predict it"

_EGUsphere, 2023_

## Author Comment (AC1)

This study by Dunkl et al. used six earth system models' output from the Decadal Climate Prediction Project to evaluate the inter-annual variability (IAV) of terrestrial gross primary production (GPP) and models' predictability of GPP IAV. Overall, the manuscript is well organized and written. The spatial patterns of GPP IAV and their predictability from environmental drivers are shown for each model. Then the differences in GPP IAV and predictability between models are clearly presented. I recommend its publication after addressing minor issues.

We thank the reviewers for their constructive and helpful comments and appreciate their careful reading of the manuscript.

After reading the text, I think implications of the findings should be addressed or discussed in-depth. Now in section 3, results are described but little linkage with findings/questions summarized in the introduction section. It is better to add more discussion and/or implications.

The reviewer is making the comment that the manuscript would benefit from putting the findings of the study into a broader perspective. This also reflects the views of reviewer No. 2. We will address this by extending the conclusions section of the manuscript to include comments on the implications of the findings and an outlook on the topic.

I am wondering whether regional results can be presented, such as Anders Ahlstorm et al., Science 2015. This would help our understanding on GPP IAV from models.

We were considering such an approach during the early stages of the analysis. However we decided that the separation of results by plant functional types would add little. As a step towards this, we will however introduce a figure showing the differences in aridity among the models. This will allow the readers to gain some insights on the contribution of biome types to GPP IAV.

---

## Author Comment (AC2)

Dunkl et al analyze 6 Earth System Models and asses the predictability of interannual variability in gross primary production on land as a function of temperature, soil moisture, and radiation.

Overall the writing in the paper is clear, and the authors describe an appropriate quantity of analysis and results. But there was very little in the way of contextualizing the results from their analysis. The authors do tie their findings frequently to the literature (see further comment #2 below), but as a reader I was left wanting to know more about why this mattered and how to go forward. These revisions can be made by the authors, although I think they are more substantial than minor revisions.

We thank the reviewers for their constructive and helpful comments and appreciate their careful reading of the manuscript. The reviewer makes a general comment expressing the need for putting the findings of the study in a broader context. This also reflects the view of Reviewer No. 1. We will address this by extending the conclusions section of the manuscript to include comments on the implications of the findings and an outlook on the topic.

Major comments:

1. It seems like there is a section missing with further discussion of implications. There is a description of the results, but then no discuss that puts these results in context. As a reader I wondered so what? What are the implications? Right now there are only a few sentences in the last paragraph of the conclusions.

We agree with the reviewer and will elaborate on this topic in the conclusions.

2. In general the results frequently appeals to other papers about why different models do or don't do something without much inclusion of those explanations here. As a reader I needed more help knowing what those previous papers had found to put these results into context. Specific examples:

A major objective of this work is to determine what is causing the differences among the analysed models. Therefore it is especially important to elaborate on these findings. We will review the manuscript for our references to these previous publications and add more details on the mechanisms that lead to the discovered differences.

line 231-232: "due to a misrepresentation of photosynthesis (O'Sullivan et al., 2020)"

This needs more explanation. Misrepresented how? I skimmed the O'Sullivan paper and I didn't find a specific "misrepresentation of photosynthesis" described, just that it matches poorly with observational products (but not *why*). This statement implies that we know why, and I'm guessing that we do not.

Here we are referring to the lines:

"*Further, the general lack of in situ observations in tropical latitudes (Cleveland et al., 2015; Schimel, Pavlick, et al., 2015), limits a realistic presentation of photosynthesis in DGVMs, impacting the TRENDYv6 GPP estimates also.*"

O'Sullivan et al., 2020 found that ESMs have a poor representation of tropical GPP because of an uncertainty in observations. These are because of a lack of flux towers in these regions and because remote sensing products rely on reanalysis data of solar radiation, which has large uncertainties in the wet tropics. We will add these arguments to our reasoning.

line 246 "complex phenological scheme"

What do you mean specifically by complex? What is different about it compared to the other models? How do you know that the performance is better because of the complexity? That evidence isn't shown here.

We agree that the manuscript would benefit from elaborating what this refers to. The cited article uses "*complex phenological scheme*" to describe models which use a plant functional type dependent parametrization of phenology. We will add this information to the manuscript.

line 307 "poor representation of soil moisture"

The paper cited (Qiao et al. 2022) uses reanalysis as their "truth" for soil moisture. Reanalysis is just another modeled product so this is a somewhat misleading statement. Better would be "poor match to other modeled products of soil moisture".

The reviewer points out that the cited study compares the performance of ESMs not with observations but with reanalysis products relying on other models. We thank for this correction and will adapt the section accordingly.

Further I don't see any obvious indication in that paper that CANESM5 is worse than other models. Soil moisture is notoriously poorly constrained by lack of observations and widely varying between models.

With our statement on soil moisture in CanESM5, we are referring to Qiao et al. 2022 Figure 8. This plot shows, that soil moisture has a low variability in CanESM5 compared to other ESMs. As the reviewer has pointed out, it does not necessarily mean a worse performance, since the uncertainty in observations does not allow to make this statement with certainty. We will correct this section to only refer to a lower variability of soil moisture in CanESM5 and the possible implications on predictability.

3. There is a lot of describing baseline climatic regions (i.e. "semi-arid" or similar) that 1) assume that readers will know what regions the authors are referring to, and 2) fail to take into account that there are background climate biases that could shift those locations across models. I suggest that the authors come up with a way to display results that allows for consistency in background climate across models (mean annual T vs mean annual P space being one option). Or that the authors show analysis that is specific to one "climate" region to demonstrate their point. Just staring at maps it was hard to translate back and forth from the statements in the text to the figures.

We thank the reviewer for this helpful insight. We will add a figure showing maps of an aridity index to the manuscript. This will not only allow to better communicate the mentioned regions to the readers but also highlight the differences among the models as the reviewer noted.

4. I found the explanation of the predictable component and predictable fraction challenging. The language wasn't hard to read, but it is a way of quantifying something that I am not familiar with and it was hard to wrap my head around what it should tell me and why. I don't have specific suggestions here but encourage the authors, who know the method well, to consider if they can make it more intuitive to a reader encountering this way of quantifying predictability for the first time. Why is this the type of predictability the authors want to know? What is the interpretation of it?

We introduced the concept of predictable component and predictable fraction to be able to describe different aspects of carbon flux predictability. This concept describes a new way to look at predictability as a multidimensional problem. We acknowledge that the introduction of this concept should take up a more prominent role in the manuscript. We will introduce a section on this issue in the introduction and add an additional subfigure to Figure 2. This subfigure will be dedicated solely to depict the difference between predictable fraction and component, and explain the need for two metrics.

Minor comments:

line 155: "For the lead years five to ten, the effects of initialization are assumed to be negligible." I assume you mean atmospheric initialization? Carbon cycle initialization would sure matter still on those time scales! Leaves probably take 20 years to equilibrate in some of these models.

Yes, we are indeed referring to climate initialization, and will adapt the text accordingly.

line 239: "GSS and GSE" There are already a lot of acronyms in this paper. I suggest just writing these out.
We are removing these acronyms.

Figure 4 and discussion of Fig. 4.

Soil moisture to what depth? Please specify. Total column? Integrated to a similar depth? Surface? Ideally you would want root-zone weighted soil moisture. It isn't mentioned how the soil moisture shown here compares to root zone.

We will elaborate on the limitations of this approach.

---

## Author Response (AR1)

In the following document, we list the additions and edits of the manuscript according to the reviewers comments. The line numbers mentioned by us refer to the track-changes file.

Reviewer No. 2:

Dunkl et al analyze 6 Earth System Models and asses the predictability of interannual variability in gross primary production on land as a function of temperature, soil moisture, and radiation.

Overall the writing in the paper is clear, and the authors describe an appropriate quantity of analysis and results. But there was very little in the way of contextualizing the results from their analysis. The authors do tie their findings frequently to the literature (see further comment #2 below), but as a reader I was left wanting to know more about why this mattered and how to go forward. These revisions can be made by the authors, although I think they are more substantial than minor revisions.

Major comments:

1. It seems like there is a section missing with further discussion of implications. There is a description of the results, but then no discuss that puts these results in context. As a reader I wondered so what? What are the implications? Right now there are only a few sentences in the last paragraph of the conclusions.

We elaborated the outlook of the study in the Conclusions section.

- Based on our findings we reason that $CO_2$ forecast skill is not an appropriate metric to evaluate the predictability of carbon fluxes and suggest an alternative metric (lines 388 – 392).

- We elaborated on our key finding that GPP variability patterns and the sensitivity of GPP to environmental drivers is directly related to the predictability of atmospheric $CO_2$ concentrations (lines 399 – 404).

- We make suggestions for the future improvement of ESM-based $CO_2$ predictability by using post-processing and utilizing the differences among the ESMs in a multi-model approach (lines 406 – 410).

- We added a final outlook on the overall limitations of the field, and the current state of the methodology in regard of these limitations (lines 411 – 414).

2. In general the results frequently appeals to other papers about why different models do or don't do something without much inclusion of those explanations here. As a reader I needed more help knowing what those previous papers had found to put these results into context. Specific examples:

line 231-232: "due to a misrepresentation of photosynthesis (O'Sullivan et al., 2020)"

This needs more explanation. Misrepresented how? I skimmed the O'Sullivan paper and I didn't find a specific "misrepresentation of photosynthesis" described, just that it matches poorly with observational products (but not *why*). This statement implies that we know why, and I'm guessing that we do not.

We explained the underlying sources for the mentioned misrepresentation of photosynthesis in tropical forests:

"While the large role of the dry tropics in driving GPP IAV is not disputed, it is likely that ESMs underestimate GPP IAV in wet tropical forests (O'Sullivan et al., 2020). This results from a limited availability of observations due to few flux towers, and because the quality of remote sensing products is limited in tropical forests due to saturation effects and a high cloud cover (Kolby Smith et al., 2016)"

(lines 244 -- 247)

line 246 "complex phenological scheme"

What do you mean specifically by complex? What is different about it compared to the other models? How do you know that the performance is better because of the complexity? That evidence isn't shown here.

We added some explanation to "complex phenological schemes" with:

"The better-performing LSMs have a high number of plant functional types or use more complex phenology schemes that depend on plant functional types and use a larger number of environmental variables to constrain phenology."

line 307 "poor representation of soil moisture"

The paper cited (Qiao et al. 2022) uses reanalysis as their "truth" for soil moisture. Reanalysis is just another modeled product so this is a somewhat misleading statement. Better would be "poor match to other modeled products of soil moisture".

We removed the evaluation of the performance of CanESM5 compared to the reanalysis data.

Further I don't see any obvious indication in that paper that CANESM5 is worse than other models. Soil moisture is notoriously poorly constrained by lack of observations and widely varying between models.

We rephrased the implications of the findings of Qiao et al. 2022 (Figure 8). Instead of evaluating the performance of CanESM5 compared to reanalysis data, we only make implications on the overall low soil moisture variability on carbon flux predictability:

"A possible explanation could be the low IAV of deep layer soil moisture in CanESM5 5 (Qiao et al., 2022). A limited ability to reproduce the full spectrum of soil moisture variability could mean that the soils have a smaller buffering capacity. As a result, they are not able to simulate the observed persistence of soil moisture anomalies, leading to a reduction of predictability."

(lines 338 – 341)

3. There is a lot of describing baseline climatic regions (i.e. "semi-arid" or similar)  that 1) assume that readers will know what regions the authors are referring to, and 2) fail to take into account that there are background climate biases that could shift those locations across models. I suggest that the authors come up with a way to display results that allows for consistency in background climate across models (mean annual T vs mean annual P space being one option). Or that the authors show analysis that is specific to one "climate" region to demonstrate their point. Just staring at maps it was hard to translate back and forth from the statements in the text to the figures.

We added a figure to show the spatial distribution of climate zoned in the analysed models (Figure 5 on page 14). We describe how the difference in climate zones explain the differences in GPP sensitivity to environmental drivers and GPP IAV (lines 301 – 311).

4. I found the explanation of the predictable component and predictable fraction challenging. The language wasn't hard to read, but it is a way of quantifying something that I am not familiar with and it was hard to wrap my head around what it should tell me and why. I don't have specific suggestions here but encourage the authors, who know the method well, to consider if they can make it more intuitive to a reader encountering this way of quantifying predictability for the first time. Why is this the type of predictability the authors want to know? What is the interpretation of it?

We rewrote the section explaining the need for two predictability metrics. In the new version, we focus more on the interpretation of the metrics, and what they can be used for, than the methodology how to calculate them. We included more examples on why the predictable fraction is not suitable to assess the predictability of $CO_2$. The new version of the methods section:

"We calculate the fraction of GPP IAV that is predictable (predictable fraction, *pf*) to assess the ability of a system to retain memory. Although is metric is useful for quantifying the mechanisms that provide predictability at a local level, *pf* is not suitable for assessing how GPP predictability allows the predictability of atmospheric $CO_2$. This is because the regions with a high *pf* do not necessarily contribute much to the global GPP fluxes. The regions with the highest *pf* are often in deserts with very little carbon fluxes (Dunkl et al., 2021). To assess the contribution of GPP predictability to atmospheric $CO_2$ predictability, we calculate the absolute portion of the IAV which can be predicted as the predictable component (*pc*). The *pc* is the difference between IAV and ensemble variability and is generally higher in regions that contribute more to $CO_2$ IAV."

(lines 201 – 209).

We also extended Figure 2 with a subfigure b) where we demonstrate the need for two metrics by applying them to two different ecosystems. The graphical representation of *pf* and *pc* in Figure 2 b) should allow a more intuitive interpretation of the metrics.

"Demonstrating the need for the two predictability metrics by the example of a tropical savanna and an arid shrubland. The predictable component (*pc*) is the absolute predictable IAV, and the predictable fraction (*pf*) is *pc* scaled by the IAV. While the arid shrubland has a better potential to retain memory (seen by the high *pf*), these ecosystems contribute little to the variability of atmospheric $CO_2$, which can be better assessed by *pc*."

(Caption Figure 2, page 9)

Minor comments:

 line 155: "For the lead years five to ten, the effects of initialization are assumed to be negligible." I assume you mean atmospheric initialization? Carbon cycle initialization would sure matter still on those time scales! Leaves probably take 20 years to equilibrate in some of these models.

The text was change accordingly:

"For the lead years five to ten, the effects of ocean and atmosphere initialization are assumed to be negligible."

(lines 156 -- 157)

 line 239: "GSS and GSE" There are already a lot of acronyms in this paper. I suggest just writing these out.

We are removing these acronyms.

(lines 256 -- 258)

 Figure 4 and discussion of Fig. 4.

Soil moisture to what depth? Please specify. Total column? Integrated to a similar depth? Surface? Ideally you would want root-zone weighted soil moisture. It isn't mentioned how the soil moisture shown here compares to root zone.

We removed the previous Figure 4 and the accompanying interpretation partially because of the mentioned limitations, and partially because the newly introduced Figure 5 showing the differences in climate zones offered a similar base to discuss climate related differences among the models.
* * *
Reviewer No. 2

This study by Dunkl et al. used six earth system models' output from the Decadal Climate Prediction Project to evaluate the inter-annual variability (IAV) of terrestrial gross primary production (GPP) and models' predictability of GPP IAV. Overall, the manuscript is well organized and written. The spatial patterns of GPP IAV and their predictability from environmental drivers are shown for each model. Then the differences in GPP IAV and predictability between models are clearly presented. I recommend its publication after addressing minor issues.

After reading the text, I think implications of the findings should be addressed or discussed in-depth. Now in section 3, results are described but little linkage with findings/questions summarized in the introduction section. It is better to add more discussion and/or implications.

This comment reflects the views of reviewer No. 2. Our answers to this comment are expressed in comment Number 1.

I am wondering whether regional results can be presented, such as Anders Ahlstorm et al., Science 2015. This would help our understanding on GPP IAV from models.

We were considering such an approach during the early stages of the analysis. However we decided that the separation of results by plant functional types would add little. As a step towards this, we added Figure 5 showing the differences in climate zones among the models. This will allow the readers to gain some insights on the contribution of biome types to GPP IAV.